# CrossTORC and WNTegration in Disease: Focus on Lymphangioleiomyomatosis

**DOI:** 10.3390/ijms22052233

**Published:** 2021-02-24

**Authors:** Jilly Frances Evans, Kseniya Obraztsova, Susan M. Lin, Vera P. Krymskaya

**Affiliations:** Division of Pulmonary, Allergy, and Critical Care Medicine, Department of Medicine, Perelman School of Medicine, University of Pennsylvania, Philadelphia, PA 19104, USA; kobra@pennmedicine.upenn.edu (K.O.); Susan.Lin@pennmedicine.upenn.edu (S.M.L.); krymskay@pennmedicine.upenn.edu (V.P.K.)

**Keywords:** mTOR, TSC1/2, Wnt/β-catenin, GSK3β, cancer, lymphangioleiomyomatosis

## Abstract

The mechanistic target of rapamycin (mTOR) and wingless-related integration site (Wnt) signal transduction networks are evolutionarily conserved mammalian growth and cellular development networks. Most cells express many of the proteins in both pathways, and this review will briefly describe only the key proteins and their intra- and extracellular crosstalk. These complex interactions will be discussed in relation to cancer development, drug resistance, and stem cell exhaustion. This review will also highlight the tumor-suppressive tuberous sclerosis complex (TSC) mutated, mTOR-hyperactive lung disease of women, lymphangioleiomyomatosis (LAM). We will summarize recent advances in the targeting of these pathways by monotherapy or combination therapy, as well as future potential treatments.

## 1. Introduction

The mechanistic target of rapamycin (mTOR) protein is a phosphatidylinositol-3-kinase (PI3) kinase-related serine/threonine kinase, which is the basis for two intracellular protein complexes, namely mTORC1 and mTORC2. When activated, these protein complexes are primarily associated with lysosomes [1,2,3]. Signaling through these complexes allows cells to coordinate intracellular catabolism, in response to a low-nutrient/high-stress extracellular milieu, and anabolism, in response to nutrient-rich, low-stress conditions [1,2,3]. Activation of mTORC1 signaling decreases autophagy and increases cellular lipid and nucleotide synthesis, messenger RNA (mRNA) translation, and protein synthesis. Overall, the mTORC1 pathway controls many aspects of metabolism, while mTORC2 activates protein kinase B (Akt) phosphorylation and regulates the actin cytoskeleton and can, in cases of mTORC1 inhibition, also activate pro-survival pathways [4].

Genetic or epigenetic changes to key mTOR pathway proteins may lead to cellular metabolic instability and human diseases, particularly cancers [5,6]. Proteins in the mTOR pathway are among the most frequently mutated genes in cancers [5,6]. In addition, hyperactive mTORC1 in stem cells can result in decreased ability to regenerate tissue and to promote aging [2]. Intracellularly, the tuberous sclerosis complex (TSC), made up of TSC1 (hamartin) and TSC2 (tuberin), is one of the major regulatory control points on the extent of activation of mTOR. In diseases in which the TSC suppressor genes are mutated, such as the female-dominant, cystic lung disease lymphangioleiomyomatosis (LAM), cell growth becomes hyperactive and uncoupled from signals from the extracellular environment [7,8,9]. LAM patients suffer from frequent lung collapses, chylous pleural effusions, abdominal tumors, and declining lung function that is worsened by pregnancy [7].

Wingless-related integration site (Wnt) signaling is a complex cell communication system that controls aspects of organ development, stem cell proliferation, injury and repair responses, and homeostatic maintenance of tissues [10,11]. Hyperactive or hypoactive Wnt signaling may result in embryonic abnormalities and/or adult organ dysfunction and diseases, in particular cancers [12]. There are multiple ligands (activating and inhibiting) and receptors and intracellular kinase-controlled complexes in the Wnt pathway that lead to either activation or inhibition of gene transcription.

During their cellular co-evolution, the highly conserved mTOR and Wnt pathways become intertwined with each other, and with other growth pathways, resulting in tangled extracellular and intracellular crosstalk. In many cancer cells, the mTOR and Wnt pathways are synergistic for growth, and when one pathway is inhibited by drug treatment, the other becomes activated.

A critical and common element of both mTOR and Wnt signaling is glycogen synthase kinase-3 (GSK3), an over-achieving serine/threonine kinase with a plethora of protein substrates and wide biological roles [13]. Rarely seen by kinases, GSK3 is active in its non-phosphorylated basal state, whereas Wnt ligand stimulation results in its phosphorylation and inhibition, which, in turn, causes activation of canonical Wnt/β-catenin signaling. Ribosomal protein S6 kinase (S6K1), a protein directly phosphorylated by mTORC1, can regulate GSK3β by mTOR-dependent feedback inhibition of Akt [14]. Inhibition of mTORC1 and S6K, by rapamycin or amino acid withdrawal, results in phosphorylation of GSK3β by Akt. Phosphorylated GSK3β is unable to activate proteosomal degradation of β-catenin, leading to its accumulation in the cytoplasm and translocation to the nucleus, where it interacts with specific transcription factors (TFs) and cofactors to stimulate cell-specific Wnt/β-catenin pathway gene transcription.

## 2. mTOR Pathway Overview

There are two mTOR complexes made up of the mTOR kinase as the central catalytic protein, with several common proteins, including the regulatory Dep domain-containing mTOR-interacting protein (Deptor), and several distinguishing proteins, namely the regulatory protein associated with mTOR (Raptor) for mTORC1 and the rapamycin-insensitive companion of mTOR (Rictor) for mTORC2. Raptor and Rictor are essential for the kinase substrate specificity of mTORC1 and mTORC2, respectively. TSC2, when complexed with TSC1, functions as a GTPase activating protein (GAP) for the small G protein Ras homolog enriched in the brain (Rheb). Rheb, in the GDP-bound form, is then unable to increase mTORC1 kinase activity. When TSC2 is inactivated, by mutation, for example in the GAP domain, or by growth factor signaling, or amino acid stimulation of Ras-Related GTPases (RAG GTPases) occurs, mTORC1 translocates to lysosomes. On lysosomal membranes Rheb–GTP activates mTORC1 [1,2] (Figure 1). 

Two of the key proteins phosphorylated by mTORC1 are S6K1 and initiation factor 4E-binding protein (4E-BP), the former promoting ribosomal synthesis and the latter releasing eukaryotic initiation factor 4E (eIF4E), the rate-limiting protein in the initiation of the translation of capped mRNAs, particularly those with a complex secondary structure. In addition, activation of mTORC1 results in increased nucleotide and lipid synthesis and inhibition of autophagy. When nutrients are scarce, another important energy-sensing regulatory kinase, AMP-activated kinase (AMPK), inhibits mTORC1. Therefore, it is not surprising that mTORC1 signaling has been linked to metabolic disease, neurodegeneration, aging, and cancer [2]. This review will focus on mTORC1 hyperactivation in cancers and LAM.

The critical role of mTORC1 in cell growth was demonstrated by the discovery that the natural product rapamycin (and subsequent rapalog mimetics) binds to the peptidyl-prolyl cis-trans isomerase FK506-binding protein 12 (FKBP12), which, in turn, binds to an allosteric site on mTOR and inhibits mTORC1 activity. Chronic treatment with rapalogs may lead to indirect secondary inhibition of mTORC2 [1,2,3]. Rapamycin (Sirolimus^TM^) was originally used as an immunosuppressant, but with the characterization of the mTOR growth pathway and its role in diseases, particularly cancer, its use widened. Unfortunately, as monotherapy in cancer, rapamycin and other rapalogs are only weakly effective [5,6]. This is not thought to be through increased drug metabolism or efflux, since rapalogs have long half-lives in vivo, but largely due to resistance due to activation of other major growth networks such as the mitogen-activated protein (MAP) kinases the receptor tyrosine kinases (e.g., epidermal growth factor (EGF), fibroblast growth factor ( FGF) vascular endothelial growth factor (VEGF)) and the Wnt/β-catenin pathways.

In contrast, in LAM, the use of rapamycin has been the difference between life and death for many women. As a result of a phenomenal collaboration between patients, clinicians, researchers, the LAM Foundation in the U.S., and worldwide LAM clinics, sirolimus was recognized as a therapy for LAM and was approved by the United States Food and Drug Administration (FDA) in 2015 for use in LAM in the U.S. [15,16]. However, some LAM patients do not respond to, or cannot tolerate, sirolimus or other rapalogs [17]. Therefore, new patient selection and treatment response biomarkers are needed, in addition to novel therapies. Our laboratory’s research is dedicated to this mission.

## 3. Wnt Pathway Overview

Wnt1, a homolog of wingless in Drosophila, was first discovered in mammalian cells as an oncogene identified through investigation of a mouse mammary tumor virus integration site. A delightful retrospective treatise including the original discovery and expansion of mammalian Wnt signaling provides an excellent introduction to this field [18]. A more recent review describes the Wnt pathway as an integral hub for developmental and oncogenic signaling networks [19]. Wnt signaling can be via both canonical GSK3β/β-catenin transcriptional signaling and non-canonical Wnt activation pathways that do not directly involve β-catenin. There are multiple non-canonical Wnt-activated pathways, with the two best characterized being the Wnt/calcium/NFAT and Wnt/RhoA/ROCK pathways [20]. However, this is a gross simplification of the tortuous paths of Wnt signaling, and in some cells, there may be crosstalk between canonical and non-canonical Wnt pathways [21,22].

Nineteen cysteine-rich glycosylated secreted Wnt family proteins can activate Wnt signaling through ten different cell surface, seven transmembrane proteins known as frizzled (FZD) receptors. These FZD receptors are associated with single transmembrane low-density lipoprotein-related proteins 5 and 6 (LRP5/6) co-receptors, which are required for Wnt/FZD signaling [21]. Wnt ligands can also bind to other cell membrane co-receptors such as the tyrosine kinases ROR and RYK [21]. In addition, there are secreted proteins such as respondins (RSPOs) and collagen triple helix related-containing 1 (CTHRC1), which bind directly or indirectly to stabilize FZD receptors [22]. Four respondin (RSPO1–4) proteins bind to the seven transmembrane proteins, leucine-rich repeat-containing GPCRs, Leucine-rich repeat-containing G protein-coupled receptors 4-6 (LGR4-6), as well as Low density Lipoprotein receptor-related Proteins 5 and 6 (LRP5/6) proteins and then amplify Wnt signaling by inhibiting ZNRF3 or RNF43, RING domain-containing ubiquitin E3 ligases that degrade FZD receptors [23,24] (Figure 2).

In the cytoplasm, newly formed β-catenin is bound to a degradation complex composed of Axin; Adenomatous Polyposis Coli (APC); casein kinase 1a (CK1a); non-phosphorylated, active GSK3β; and an E3 ligase. In the absence of Wnt proteins, β-catenin is targeted for proteolysis through phosphorylation by CK1a and GSK3β. In the presence of canonical Wnt ligands, the scaffold protein Disheveled (DVL) is activated, which induces the dissociation of the β-catenin–ubiquitin E3 ligase degradation complex and non-phosphorylated β-catenin. β-catenin accumulates in the cytoplasm and translocates to the nucleus [2,12,20]. Nuclear β-catenin binds to and activates T cell factor (TCF) and lymphoid enhancer-binding (LEF) TFs, which directly bind to specific DNA sequences, enhancing the transcription of genes such as Cyclin D, c-Myc, and Axin2 and inhibiting transcription of other genes such as E-cadherin and the tumor suppressor p53 [19,25]. There are many other transcriptional cofactors that bind to β-catenin and play a critical role in tissue-specific outcomes of Wnt/β-catenin signaling [25].

Further complicating any analysis of canonical Wnt/β-catenin signaling is that particularly in epithelial cells, β-catenin binds to cadherin on the plasma membrane at the cell adhesion junctions [26]. While the half-life of the transcriptional signaling pool of β-catenin is in the order of minutes, the membrane junction pool is quite stable. It is thought that the membrane-adhesive and nuclear transcription roles of β-catenin may be independent [27]. However, this relationship has still not been clearly defined [11].

The non-canonical Wnt-activated pathways that do not activate FZD receptors or directly stabilize β-catenin are less well studied [20]. The best-characterized ones are the calcium NFAT and the planarity Rho ROCK kinase pathways, which are known to alter cellular patterning and re-organization [28,29]. Both canonical and non-canonical Wnt pathways may be active in the same cell, and non-canonical Wnt signaling may induce nuclear translocation of β-catenin. This review will consider only the canonical Wnt/β-catenin pathway. To further complicate the control of the Wnt pathway, there are also five families of soluble and extracellular Wnt antagonistic proteins, including the Dickkopf family, which bind to LRP5/6 and prevent Wnt binding [30].

## 4. mTOR and Wnt Pathway Crosstalk

The mTOR and Wnt cancer cell signaling pathways, and indeed all major mammalian cell growth pathways, have diverse extracellular activating and inhibiting ligands and multiple points of intersection. Intracellular crosstalk between these pathways is complex and has been demonstrated in many cells. The mTOR and Wnt signal transduction mechanisms were initially discovered over 30 years ago. Since then, there has been somewhat confusing literature on the crosstalk between these important growth and developmental pathways. Resistance to pathway inhibitors has mainly been related to activation/inhibition by phosphorylation and dephosphorylation of key proteins under specific in vitro/in vivo conditions. In the majority of cancer cells, stimulation of the Wnt/β-catenin pathway directly activates the transcription of genes coding for specific growth and proliferative mRNAs. Activation of the mTOR pathway, via many growth factors, importantly including the insulin growth factor family, results in increased translation, particularly of capped mRNAs, with complex 5-prime structures, including Wnt-activated genes such as c-Myc.

Many growth networks are cooperative in cancer cells, and when one is inhibited or mutated, the other is activated, resulting in resistance and continued growth. An example of cooperative growth pathways for mTOR and Wnt was demonstrated in breast cancer cells in transgenic mice overexpressing the glycoprotein Nmb, having increased β-catenin transcriptional activity and phosphorylation of mTOR and 4E-BP1 [31]. An example of reciprocal control between PI3K/Akt/mTOR and Wnt/β-catenin networks has been shown in colorectal cancer (CRC), where inhibition of one pathway activates the other [32]. In prostate cancer, PI3K/Akt/mTOR/AR/MAPK/Wnt pathways cooperate to enhance cancer growth and drug resistance [33]. In liver cancer, levels of p-mTOR-S2448, an indicator of mTORC1 activation, were dramatically elevated in tumors with β-catenin mutations and glutamine synthetase upregulation [34]. However, in some cases, the inhibition of one pathway results in antagonism of the other. For example, AMPK activators, which decrease mTOR activity and increase autophagy, suppress cervical cancer cell growth by impairing protein synthesis of DVL3, a positive regulator in Wnt/β-catenin signaling [35]. In addition, in some multiple myelomas, AMPK was activated by β-catenin silencing, resulting in decreased levels of activated mTOR [36].

In stem cells, hyperactive mTORC1 generally inhibits the Wnt/β-catenin pathway, resulting in stem cell exhaustion [37]. mTORC1 activation was shown to decrease canonical Wnt/β-catenin signaling in specific stem cells by increased degradation of Wnt FZD receptors [38]. These authors hypothesized that sustained mTORC1 signaling in stem cells results in a decreased stem cell number and ultimately decreases life span. This agrees with the effect of rapamycin on the extension of the life span in mice and reflects a balance between self-renewal and differentiation for long-term maintenance of regenerating stem cells [2,37]. Antagonistic crosstalk has also been demonstrated in TSC1/2-mutated melanocytes and melanomas, with hyperactive mTORC1, in which active β-catenin was reduced and associated with reduced pigment gene expression [39]. These authors suggested that loss of TSC suppressors upon GSK3/β-catenin activation can have opposing effects, dependent on specific cell types and external conditions.

There are multiple ways a cell can respond to stimulation of canonical Wnt and mTORC1 pathways. We highlight this in a highly simplified cartoon, the Wnt/β-catenin activation of transcriptional events and mTORC1 activation of translational events (Figure 3). This results in either synergistic growth or differentiation, or a combination of effects, or, indeed, even dedifferentiation [37]. Many more sophisticated controls on these pathways are the result of millions of years of evolution.

In both TSC and sporadic (non-inherited) LAM patients, mutations in TSC2 have been related to increased active β-catenin in kidney angiomyolipomas and pulmonary LAM cells [40,41]. An increase in β-catenin in LAM lung biopsy tissues has also been strongly associated with the melanocyte/LAM protein HMB45 and smooth muscle α-actin and suggested to be a specific biomarker for LAM cells [42,43]. However, this was unable to be investigated in vivo since LAM lung cells have not been able to be grown in mice [7,8]. Wnt pathway genes were also observed in LAM-specific mesenchymal lung cells by two important single-cell RNA sequencing (scRNAseq) studies in LAM [44,45].

Our research showed upregulation of the Wnt pathway in LAM-lung-specific mesenchymal cells [45]. To follow this up in vivo, we deleted the *Tsc2* gene, specifically in mouse lung mesenchymal progenitor cells, resulting in progressive age-dependent enlargement of alveolar spaces, as well as pulmonary lung function decline [45]. Selected genes from the Wnt pathway, including *Wnt3a, Wnt4, Wnt5a*, and *Wnt10b*, encoding for Wnt ligands, *Fzd1, Fzd2*, and *Fzd8*, encoding for FZD receptors, were elevated in the *Tsc2*-knockout lung mesenchyme, while *Wnt3a* and *Wnt5a* gene expression appeared to be downregulated by in vivo rapamycin treatment. Rapamycin, however, did not reverse but slowed down the alveolar space enlargement phenotype. We highlighted elevated crosstalk between *Tsc2*-deficient mesenchymal-derived cells and surrounding epithelial cells via canonical Wnt ligand/receptor signaling [45]. Genetic inactivation of Wnt signaling by deletion of the β-catenin-encoding gene (*cttnb1*), in addition to mTORC1 activation in the mouse lung mesenchyme, reversed age-dependent alveolar enlargement. However, Wnt pathway activation by genetically stabilized β-catenin in the lung mesenchyme was not sufficient for the development of the observed phenotype in this model [45]. Together, these data suggested that activation of both mTORC1 and canonical Wnt/β-catenin pathways act synergistically in triggering the development of the LAM-like phenotype.

LAM occurs almost exclusively in women and is exacerbated by pregnancies. Consistent with these observations of LAM susceptibility, the most severe lung structure changes and lung function decline was observed in lung mesenchymal *Tsc2*-deficient breeding females. We found that select genes, including *Wnt3a* and *Wnt5a*, were differentially upregulated specifically in females, suggesting the involvement of estrogen-dependent regulation. Indeed, the stimulating effect of estrogen on *Wnt3a* and *Wnt5a* was reported in earlier studies [46]. Additionally, these genes were found among estrogen receptor α (ERα) transcriptional targets [47]. It is of note that expression of the ERα was elevated in *Tsc2*-null mouse lung fibroblasts. Besides the activation by estrogen, ERα is activated via phosphorylation by S6K1, an enzyme that is directly activated and phosphorylated by mTORC1 [48] (Figure 2). These data suggest that the interplay of all three pathways, namely mTORC1, Wnt/β-catenin, and ERα, is involved in LAM lung phenotype development. An additional connection between these pathways is GSK3β, which can be inactivated by estrogen treatment via phosphorylation, leading to β-catenin stabilization and subsequent activation of the Wnt/β-catenin pathway [49] (Figure 4).

## 5. Therapeutics for mTOR and Wnt Pathways

As the knowledge of proteins central to mTOR and Wnt signaling was gained, some of these proteins were targeted for inhibition or activation and investigated as potential drug targets. Unfortunately, the essentiality of these networks, and the exquisite cellular evolution of proteins to modulate them, has largely prevented the successful development of drugs. At the date of writing this review, there were only three rapalogs and no marketed dual mTOR inhibitors or specific Wnt modulator drugs, but multiple therapies are in preclinical and early clinical development. We will discuss mTORC1 pathway inhibitors, followed by Wnt pathway inhibitors and then combinations thereof.

## 6. mTOR Pathway Inhibitors

Rapamycin (sirolimus) is the original mTOR pathway inhibitor, as discussed previously. This allosteric mTOR inhibitor binds specifically to FKBP12 protein, which, in turn, binds to a site on mTOR in context with the mTORC1 complex. Sirolimus has long-lasting pharmacokinetics and a reasonable toxicity profile at 1–2 mg daily [50]. This profile, and efficacy in halting lung function decline, has made rapamycin a lifesaver for many women with LAM [15,16]. However, higher concentrations of rapamycin may result in drug resistance. For example, in a Tsc2-null, rapamycin-resistant cell line, which was highly tumorigenic in mice, β-catenin signaling was shown to be upregulated [51].

Many close analogs of sirolimus have been developed for clinical use, including everolimus (RAD001) and temsirolimus [52]. These so-called rapalogs were designed by the addition of chemical groups to the natural product sirolimus, but they each have different pharmacokinetic and pharmacodynamic properties. Direct ATP-competitive mTOR kinase inhibitors, such as Torin1/2, MLN0128, CC-223, Sapanisertib, and AZD2014 (Vistusertib), are effective in preclinical studies, and some in phase I/II early clinical trials have not had safety profiles that have allowed them to progress into phase III. A group of compounds, named Rapa-Links, have been made by the addition of a direct kinase inhibitor to rapamycin been tested in the clinic [52].

There are many PI3K/mTOR mixed inhibitors, but only a few have advanced in clinical cancer trials due to pharmacokinetic, tolerability, and safety issues. An example of a promising PI3K/mTOR inhibitor with phase 1 clinical data in idiopathic pulmonary fibrosis is GSK2126458 (omipalisib) [53]. In preclinical in vitro studies in *Tsc2*-null cells, we showed that omipalisib inhibited phosphorylation of Akt and both protein synthesis arms downstream of mTORC1 activation, namely S6K and 4EBP1 [54]. It is not widely appreciated that rapamycin, and other allosteric rapalogs, inhibit phosphorylation of S6K at much lower concentrations than they inhibit phosphorylation of 4E-BP1 [55]. Rapamycin, at clinically tolerable concentrations of 5–15 nM [50], mainly inhibits S6K/S6 but not 4E-BP1/eIF4E, resulting in feedback inhibition of GSK3β [14].

Another important kinase sensor of the extracellular environment growth conditions is AMPK, which phosphorylates and activates TSC2 suppression of mTOR, and one clinically available AMPK inhibitor metformin has been repurposed for cancer treatments [56]. However, it should be noted that metformin has multiple cellular targets, all of which may contribute to its clinical effectiveness.

Excellent basic research on the mechanism of growth of LAM cells has resulted in repurposing several older drugs for LAM. To date, only phase I clinical tolerability studies, including with the cyclooxygenase-2 inhibitor celecoxib, the HMGCoA reductase inhibitor simvastatin, and the autophagy inhibitor hydroxychloroquine, have been completed [57,58]. A series of publications have shown that the natural product modified nucleoside analog mizoribine is a potent inhibitor of LAM growth in vitro and in vivo through inhibition of inosine monophosphate dehydrogenase (IMPDH), an enzyme that synthesizes guanylate nucleotides, which become limiting in cells with hyperactive mTORC1 [59]. Unfortunately, due to its mechanism of action, mizoribine is not additive for growth inhibition with rapamycin.

## 7. Wnt Pathway Inhibitors

No specific Wnt pathway inhibitors have been clinically approved. However, some multi-target inhibitors, such as lithium chloride via inhibition of GSK3-β or inhibitors of cyclooxygenase-2, a gene upregulated by β-catenin, have shown benefits, which may be partially due to the Wnt/β-catenin pathway inhibition [60]. Such inhibition can occur at multiple targets, including, but not limited to, an extracellular block of various Wnt ligands, direct antagonism of FZD receptors on the cell membrane, intracellular inhibition of Wnt palmitoylation by PORCN, or nuclear inhibition of β-catenin transcription [12,61]. Several monoclonal antibodies (mAbs) have been developed to inhibit the Wnt pathway, including the mAb OMP-131R10 (rosmantuzumab) that blocks RSPO3 or OMP18R5 (vanticumab) that blocks FZD receptors [61]. Given that a low concentration of nuclear β-catenin may be sufficient for active target gene transcription, inhibition of the ligands or receptors or cytoplasmic Wnt modulators may not be as effective as antagonism of the binding of β-catenin to nuclear transcription activators of proliferation [62].

β-Catenin requires many transcriptional co-activators to generate a transcriptionally activated complex, including cyclic AMP response element-binding protein (CBP) and histone acetyltransferase EIA-binding protein p300. It has been shown that genes transcribed from the CBP/β-catenin interaction are mainly proliferative, whereas those transcribed from the p300/β-catenin interaction are mainly differentiation genes [63]. ICG-001 and its more potent analog PRI-724 (active compound C82) disrupt the interaction between CBP/β-catenin but not p300/β-catenin [64]. The phosphorylated PRI-724 is rapidly converted in vivo to the non-phosphorylated form C82. PRI-724 has been in phase 1 cancer and fibrosis clinical trials NCT01302405 and NCT01764477 [65,66], but this drug requires intravenous delivery for systemic exposure and has some hepatic toxicities. Topical addition of the drug was safe and beneficially changed skin biomarkers but did not show efficacy in a short clinical trial on systemic sclerosis [67]. A pulmonary target may allow local, rather than systemic, delivery of such drugs. However, C82 has not been used locally in the lung but is a useful compound to interrogate the interaction of mTORC1/Wnt/β-catenin pathways in LAM and Tsc2-null cell in vitro studies.

An intriguing link between mTOR and Wnt pathways is the mTOR complex protein Deptor that has been shown to be a tumor promoter in CRC and a downstream target of canonical Wnt and c-Myc signaling [68]. This is somewhat surprising since Deptor binds to, and inhibits, mTORC1 activity [1,2,3]. The β-catenin pathway inhibitor ICG-001 increased pS6 and pAkt in some CRC cells, suggesting a differential effect on mTORC1 [68]. The authors suggest that the knockdown of Deptor may induce differentiation and inhibition of CRC cells. However, in these studies, mTOR inhibition enhanced the antitumor activity of ICG-001, highlighting the potential clinical effectiveness of combined inhibition of mTOR and Wnt.

## 8. Targeting Multiple Growth Pathways

In many oncology indications, targeting a single growth pathway by monotherapy usually results in therapeutic resistance by compensatory upregulation of the signaling of other growth pathways. Therefore, combination therapies may result in additive or synergistic inhibition of tumor growth. Several preclinical studies with combination mTOR/Wnt therapies have shown efficacy. For example, dual PORCN inhibitor and PI3K/mTOR inhibitor treatment showed added efficacy in Wnt-driven pancreatic cancer models and RSPO3-translocated cholangiocarcinoma cells [69]. In mice transplanted with acute myeloid leukemia, a combination of a dual PI3K/mTOR inhibitor (VS-5584) and a Wnt inhibitor (ICG-001) acted together synergistically to reduce the leukemic burden and prolonged survival in mice with high PRL-3 phosphatase [70]. However, while these preclinical studies are encouraging, we are aware of the difficulties in clinically titrating combination therapies to limit toxicities [71]. Indeed, each drug may alter the other’s bioavailability, requiring careful pharmacokinetic and pharmacodynamic profiling in early-phase clinical trials. The mechanistic signaling pathways in each subtype of cancer require incredibly complex algorithms to determine the best therapeutic approaches. For example, to overcome resistance to non-small-cell lung cancer monotherapies, a novel therapeutic interventional mapping system and algorithm was developed, based on the integration of genomic and transcriptomic data, that may allow customized combinations of therapy [72].

As outlined above, autocrine and paracrine interaction between the mTOR and Wnt/β-catenin pathways can result in complex crosstalk that is dependent on specific cell types and external conditions. Therefore, to support cross-pathway regulation, in a specific setting, there need to be multiple lines of evidence. One of these methods is in vitro and in vivo genetic ablation or overexpression, and another is the use of specific pathway modulators. As shown by our research, in our in vivo lung mesenchymal progenitor *Tsc2*-knockout mouse model, the genetic deletion of β-catenin also in mesenchymal lung cell progenitors prevented the alveolar structural changes induced by hyperactivation of mTORC1 in these cells [45].

To follow the clinical profile of new therapies, it is critical to identify non-invasive biomarkers reflecting both the specific pathway modulation and the relationship to disease progression. For LAM, the only recommended non-invasive biomarker is VEGF-D [73]. Treatment of LAM patients with sirolimus has been shown to decrease VEGF-D [74]. Many other biomarkers in LAM patients have not shown as clear a relationship as VEGF-D, but studies continue to attempt to match lung function changes with biomarkers [75]. Another potential non-invasive biomarker for LAM is the carbohydrate-binding secreted protein Galectin-3 (Gal-3) [76,77,78]. Preclinical studies showed that in global mesenchymal progenitor *Tsc2* knockout tumors, with vascular anomalies, Gal-3 was overexpressed [77], and in Gal-3 knockout mice, active β-catenin was reduced in the lung.

A difficult challenge for scientists and clinicians is to design clinical trials in an orphan disease with a small number of patients. Using the patients as their own control, or adaptive designs, as is done in many cancer trials, may give the most meaningful data and require fewer patients [79,80]. Overall, a better understanding of specific disease drivers, and their measurable biomarkers, and the spatial and temporal balance between growth and differentiation in subsets of cells may allow a better prediction of patients who will respond best to a particular therapy and allow more individualized therapy.

## 9. Conclusions and Potential Future Therapeutics

The mTOR and Wnt pathways have evolved together, forming convoluted crosstalk both intracellularly and extracellularly. The great number of nodes controlled by phosphorylation and dephosphorylation provides a potential plethora of drug targets. However, the exquisite crosstalk between cellular growth pathways makes it unlikely to be able to inhibit any one kinase without encountering feedback mechanisms. For example, inhibition of S6K1, a key substrate of mTORC1, results in activation of the kinase Akt. In addition, while it has been possible to design selective ATP-competitive kinase inhibitors, many of them have significant pharmacokinetic, tolerability, and safety issues. This is an age-old problem for medicinal chemists and biologists [81].

One of the ways to speed up the discovery of selective compounds with good metabolic profiles is to use artificial intelligence algorithms that use the chemical shape of small molecules both as to fit and selectivity with the target and a lack of shape characteristics that result in poor pharmacokinetics and toxicity [82]. Another way to use the cells’ protein degradation system to advantage is to couple a target-selective small-molecule compound to a ubiquitin E3 ligase module [83]. These so-called proteolysis targeting chimeras (PROTAC) compounds result in the protein target being ubiquitinated and directed to the proteasome for degradation. When the target is a kinase, the PROTAC compound may have two effects, both inhibition of the kinase activity and degradation of the kinase. However, many PROTAC compounds have been designed around the immunomodulatory imid (IMID) backbone structure of thalidomide and therefore may have some of the potential safety issues related to this class of drugs. As with all novel therapies, unknown benefits and toxicities need to be further studied.

Quantum advances in mAb discovery and development have occurred before and during the Corona Virus Disease (COVID-19) era [84]. These include improvements and decreases in costs of production, novel advances in combination antibody therapies, and DNA-directed [85] and mRNA-directed antibody production [86]. To date, there are over 30 mAb therapies approved by the FDA for cancer indications, and this number is sure to rise in the next few years. The development of immune checkpoint inhibitors, such as anti- programmed death ligand 1 (PD-L1) antibodies, has led to preclinical studies in mTOR-driven diseases, such as LAM [87,88], and combinations with mTOR/PI3K inhibitors in cancer [89]. Antibody–drug conjugates (ADCs) can also be used to target drugs to cells overexpressing the protein that the antibody recognizes. An interesting example of this is PD-L1-Dox, the conjugation of a PD-L1 antibody to doxorubicin, which showed in vivo efficacy in a breast cancer model [90]. Another example of ADC targeting the Wnt pathway is the specific LGR5 mAb conjugated to the cytotoxic drug auristatin, which showed efficacy in vivo in a xenograft colon cancer model [91]. To date, there are eight FDA-approved ADCs for cancer [90]. For example, with a better understanding of specific cell surface proteins, it may be possible in the future to use an ADC approach to target rapamycin, or another drug, specifically to the LAM-specific cells identified by scRNAseq [42,43]. However, despite improvements in production and development, novel antibody therapy remains out of reach for many patients around the world.

New therapeutic opportunities involve the modification of DNA or RNA targets. Genome editing, particularly by CRISPR/Cas9 technologies, holds promise for ex vivo and in vivo genome editing of diseased genes, but still, many hurdles need to be cleared to enhance the safety of these therapies [92]. Another therapeutic opportunity could be to reduce long non-coding RNA oncogenes such as gastric carcinoma proliferation enhancing transcript 1 (GHET1), which has been shown to activate both mTOR and Wnt/β-catenin pathways in cervical cancer [93]. The expanding universe of microRNAs (miRNAs), and their association with diseases, also provides more potential anti-cancer targets. Rapamycin has been shown to increase miR-29b, which promotes mTORC1-hyperactive growth in TSC2-deficient cells via downregulation of the TS retinoic acid receptor β, or reduction in miRNAs, such as miR-29b itself, to increase tumor suppressor pathways [94]. Perhaps most exciting would be the advances in stem cell research that would allow the regeneration of lung tissue or, indeed, the generation of the new adult lung [95,96,97]. However, since mTOR is central to cell-specific, spatial, and temporal control of tissue repair and regeneration, manipulation of this pathway will remain a major challenge for medical research [98].

Many of these novel therapies will require sizable investments to develop. Therefore, for an orphan disease like LAM, the pragmatic, mechanism-driven repurposing of approved drugs may provide more immediate clinical benefit. As has been quoted many times before, the best way to discover a new drug is to start with an old one. However, it is necessary to note that many of the older drugs are less selective than more recent compounds, so understanding their mechanism of action, correct clinical doses, and possible toxicities in different patient populations will be critical for therapeutic success.

Diseases, particularly cancers, which result from dysfunction in both mTOR and Wnt pathways, await breakthroughs in new compounds, antibodies, nucleic acid drugs, and bioengineering advances for the clinical development of the next generation of therapeutics. We conclude with a touch of levity that *CrossTORC* and *Wntegration* of convoluted cell signaling cascades increase the challenges of finding safe and effective therapeutics and will require productive collaborations between patients, scientists, clinicians, and the pharmaceutical industry. A shining example of such a collaboration is in the orphan disease LAM, where such a cooperative community partnership resulted in the development of sirolimus as an approved therapy for LAM.

## Figures and Tables

**Figure 1 ijms-22-02233-f001:**
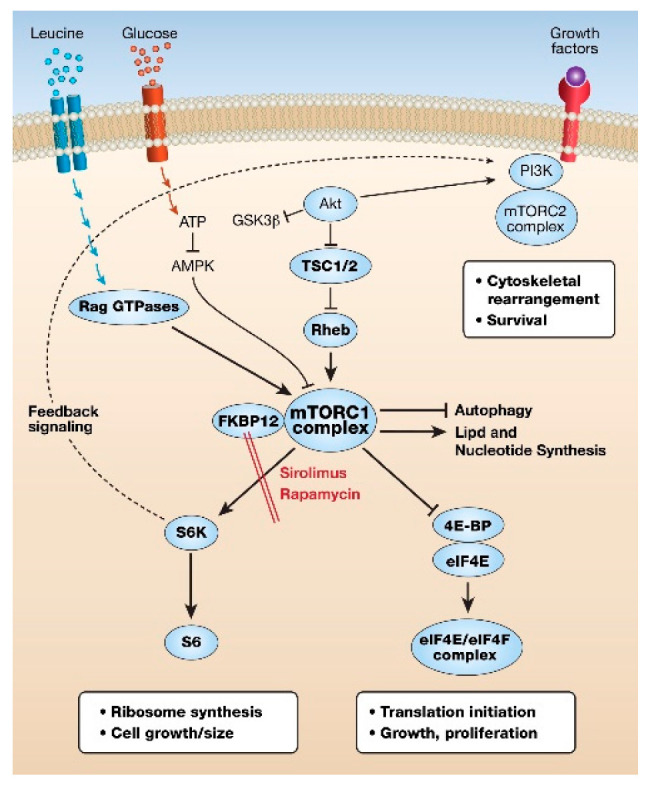
Cartoon of major proteins in the mechanistic target of rapamycin (mTOR) growth pathway. Extracellular high-energy conditions decrease the suppressor activity of the tuberous sclerosis complex (TSC)1/TSC2 and amino acids activate RAG GTPases and Ras homolog enriched in the brain (Rheb) GTP-binding protein, stimulating mTORC1 kinase. mTORC1 phosphorylation activates S6 kinase (S6K) and S6 ribosome synthesis. Phosphorylation of eukaryotic initiation factor 4E (eIF4E)-binding protein (4E-BP) releases the rate-limiting initiation factor eIF4E, resulting in enhancement of the translation of complex capped mRNAs. Overall, increased mRNA translation, increased lipid and nucleotide synthesis, and inhibition of autophagy result in cellular growth and proliferation.

**Figure 2 ijms-22-02233-f002:**
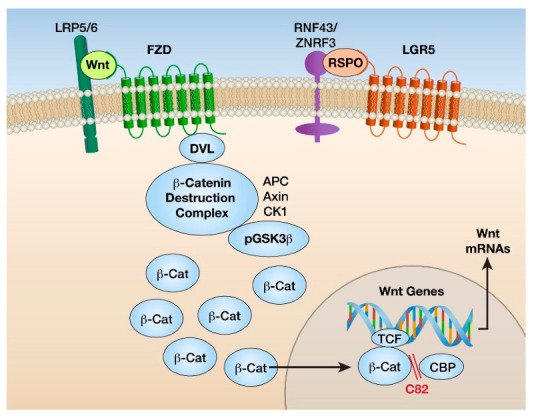
Cartoon of major proteins in the wingless-related integration site (Wnt)/β-catenin growth pathway. Wnt ligands bind frizzled (FZD) and LRP5/6 cell surface receptors. FZD receptors are stabilized by respondin (RSPO) binding to LGR5 and inhibition of ubiquitin E3 ligases (ZNRF43, ZNR43). Intracellularly, the Disheveled (DVL) protein binds to FZD and inhibits the β-catenin destruction complex and CSK3β, resulting in accumulation of non-phosphorylated β-catenin in the cytoplasm. β-catenin translocates to the nucleus and interacts with the transcription T cell factor (TCF) and the transcription cofactor cyclic AMP-binding protein (CBP). This results in the transcription of Wnt-specific genes. The inhibitor C82, the active form of PRI-724, prevents β-catenin from binding to CBP and inhibits Wnt transcription.

**Figure 3 ijms-22-02233-f003:**
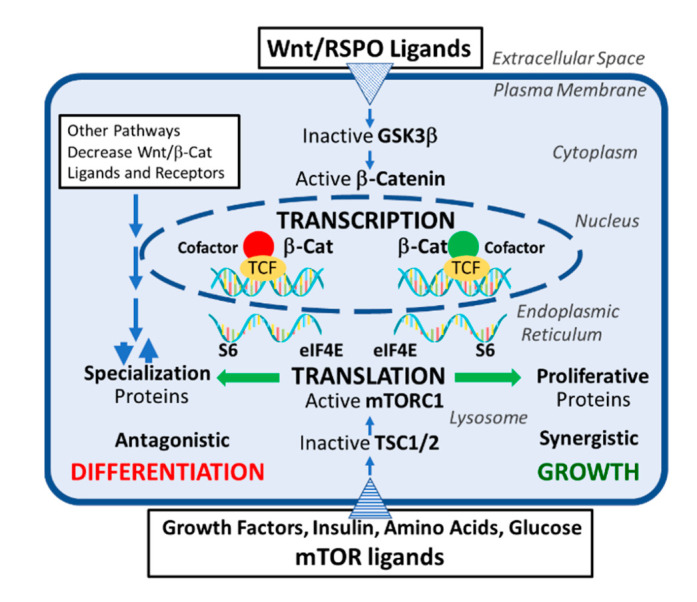
Simplified cartoon of potential crosstalk between activated canonical Wnt/β-catenin and mTORC1 pathways. Wnt ligand activation of Frizzled (FZD) plasma membrane receptors results in phosphorylation and inactivation of glycogen synthase kinase-3 beta (GSK3β), which can no longer phosphorylate β-catenin for destruction. β-catenin accumulates in the cytoplasm and translocates to the nucleus, where it binds with T cell factor (TCF) transcription factors and other cell-specific cofactors. The relative amount of differentiated (red circle) vs. growth (green circle) β-catenin-binding cofactors results in transcription of mRNAs for differentiation or proliferative proteins. Other signaling pathways, such as the mitogen-activated protein (MAP) kinase pathway, may alter Wnt/β-catenin/mTORC1 crosstalk.

**Figure 4 ijms-22-02233-f004:**
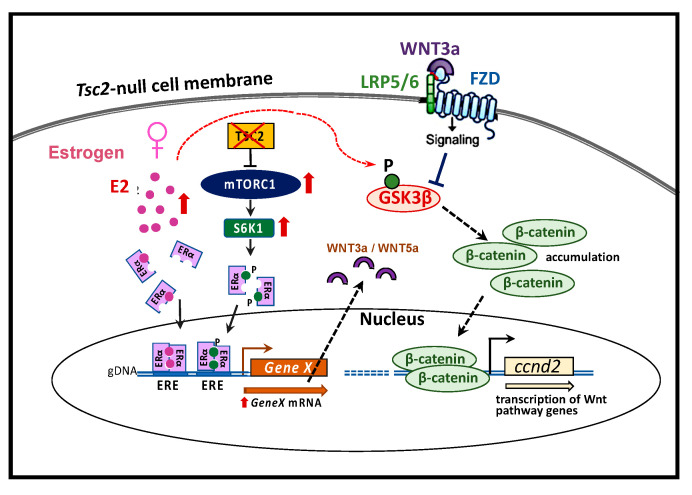
Cartoon of mTORC1, Wnt/β-catenin, and estrogen receptor α (ERα) molecular pathway interplay in the female-specific, Tsc2-null, rare lung disease lymphangioleiomyomatosis (LAM). Estrogen activates ERα nuclear localization and dimerization. In the absence of Tsc2 suppressor function, mTORC1 activates S6K phosphorylation, which, in turn, phosphorylates and activates ERα. Estrogen-stimulated genes, such as the Wnt ligands Wnt3a/5a, are transcribed and can in an autocrine manner activate frizzled (FZD) and LRP5/6 Wnt receptors. S6K also phosphorylates GSK3β, resulting in deactivation of the β-catenin destruction complex. β-catenin translocates to the nucleus, enhancing transcription of Wnt/β-catenin specific genes, such as cyclin D2 (ccnd2).

## Data Availability

Not applicable.

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
