# Peer review of "CrossTORC and WNTegration in Disease: Focus on Lymphangioleiomyomatosis"

_ijms, 2021, doi:10.3390/ijms22052233_

Round 1
Reviewer 1 Report
This review manuscript gathers many detailed information about mTOR and Wnt signaling pathways in LAM. It comprises status of research and also summarizes clinical translation. Potential novel treatments are described giving an outlook into future therapeutic options.
Major comments
The paragraphs 2 and 3 are hard to read without a figure showing the actual signaling cascades. It would be good to refer to one of the figures (maybe expanded) in the manuscript or to add another figure that visualizes the signaling cascades.
The authors refer to LAM as exemplifying disease for the relevance of mTOR/Wnt signaling pathway crosstalk. A few lines should describe the nature and symptoms of this disease.
When describing their own research, it is not clear whether the authors used LAM patient cells or cells from a LAM model (Line 212). Also, it is not common knowledge how a LAM model is generated.
There are a few ATP-competitive mTOR inhibitors that reached clinical development. Please mention those.
Minor Comments
- Throughout the manuscript it is not clear what the difference/function of GSK and GSK3b is. Both terms are used intermittently.
- L. 59 should be: phosphorylates GSK3b by AKT
- L. 87: FKB12 or FKBP12? Explain abbreviation.
- L93/94: Please re-phrase and also mention other resistance mechanisms
- L156: please re-phrase
- L. 164: should be: Cancer cell stimulation
- L. 170: resistance "to pathway inhibitors"
- L. 209 should be "result", delete s
- L. 222 should be selected genes
- L. 266-269: Please re-phrase
- L. 276 should be "RAD001"
- L 332: systemic sclerosis or sclerotic?
- L. 333: Has CC82 been used locally in the lung?
- L. 345: add: in many oncology indications, targeting a single.....
- L. 399: could mean: .... without encountering feedback activation mechanisms
Author Response
Reviewer 1 Major comments
1.Paragraphs 2 and 3 are hard to read without a figure of the signaling cascades.
We have added 2 figures, Figure 1 for the mTOR pathway and Figure 2 for the Wnt/b-catenin pathway.
2.Add a few lines to describe the nature and symptoms of LAM disease.
We have added a sentence at now Line 39,40: LAM patients suffer from frequent lung collapses, chylous pleural effusions, abdominal tumors and declining lung function that is worsened by pregnancy [7].
3.Previous line 212 are these LAM patient cells or cells from a LAM model? Also, it is not common knowledge how a LAM model is generated.
We added the information that human LAM lung cells cannot be grown in vivo. Also, we added that our paper reference 45 first showed Wnt pathway upregulation in human LAM-lung-specific mesenchymal cells, then we followed this by a Tsc2 knockout specifically in mouse lung mesenchymal progenitor cells.
[42,43]. However, this was unable to be investigated in vivo since LAM lung cells have not been able to be grown in mice [7,8].
Our research showed up-regulation of the Wnt pathway in human LAM-lung-specific mesenchymal cells [45]. To follow this up in vivo we deleted the Tsc2 gene, specifically in mouse lung mesenchymal progenitor cells…
- There are a few ATP-competitive inhibitors in clinical trials. Please mention those.
We discovered that most ATP-competitive mTOR inhibitors are not continuing in clinical trials. However, to clarify this we added line 324: Direct ATP-competitive mTOR kinase inhibitors, such as Torin1/2, MLN0128, CC-223, Sapanisertib and AZD2014 (Vistusertib), are effective in preclinical studies, and some into Phase I/II early clinical trials none have had safety profiles that have allowed them to progress into Phase III.
Reviewer 1 (continued)
Minor comments:
- GSK and GSK3b are used interchangeably.
We followed notation as given in the references related to each study.
- L59 changed to ‘results in phosphorylation of GSK3b by Akt” for clarity. New line #671
- L87 FKB12 or FKBP12?
Thank you for correction. We had missed out the P here. We write out the full name with explanation New line #102. the peptidyl-prolyl cis-trans isomerase FK506-binding protein 12 (FKBP12)”
- L93/94 Rephrase and mention other resistance mechanisms.
New line # 109 “This is not thought to be through increased drug metabolism or efflux, since rapalogs have long half-lives in vivo, but largely due to resistance due to activation of other major growth networks such as the MAP kinase (Ras, Raf, MEK, ERK), the receptor tyrosine kinase (eg EGF, FGF, PDGF) and the Wnt/b-catenin pathways.”
- L156 Rephrase
New line # 180 Both canonical and non-canonical Wnt pathways may be active in the same cell, and non-canonical Wnt signaling may induce nuclear translocation of β-catenin. This review will consider only the canonical Wnt/b-catenin pathway.
- L164 should be Cancer cell stimulation. Change New Line #189 “The mTOR and Wnt cancer cell signaling pathways…”
- L170 should be resistance to pathway inhibitors. New Line #195 ‘Resistance to pathway inhibitors”
- L209 should be result delete the s. New Line 234 altered phrase to remove can, so results is correct.
- Line 222 Should be selected genes. New Line 259 ‘This results’
- Line 266-269 rephrase. We deleted this phrase, New Line #310 since it is discussed again later.
- Line 276 should be RAD001. Correction made New Line # 321.
- Line 332. Systemic sclerotic should be sclerosis. Correction made New Line # 379.
- Line 333. Has C82 been used locally in the lung. Comment added ‘However, C82 has not been used locally in the lung’ New Line # 380.
- Line 345. Add “In many oncology indications, targeting a single.’
- Line 399 rephrase. Rephrased at New Line # 448 ‘to inhibit any one kinase without encountering feedback mechanisms.”

Reviewer 2 Report
The authors have provided a broad overview of the two central human cell signaling pathways, namely mTOR and wnt, in the context of the rare lung disease, lymphangioleiomyomatosis (LAM). The manuscript is well written, balanced, and highlights the inherent difficulties/pitfalls in targeting key central pathways for therapeutics.
I have only a few very minor edits for the authors consideration:
1. Abstract: Change mammalian to mechanistic
2. Line 34: change tumor to tuberous
3. Line 187 and beyond: replace SCs with stem cells
4. Line 190: remove the word stem
5. Line 236: Odd construction of the sentence. Please revise.
6. Line 300: mention other completed trials: statins, hydroxychloroquine
7. Line 332: change to sclerosis
8. Lines 377 - 386: Don't add much to the review, are a bit tangential, and may be biased by the reported COI. Suggest removing.
Author Response
Please see attachment and:
Reviewer 2 comments
1.Abstract: changed mammalian for mechanistic.
- Line 34: changed tumor for tuberous.
- Line 187 and beyond (now line 217 and beyond) relaced SCs with stem cells
- Line 190 (now line 223) removed the word stem
- Line 236 (now line 271) Odd construction of sentence. Changed the sentence into 2 sentences:” LAM disease occurs almost exclusively in women and is exacerbated by pregnancies. Consistent with these observations of LAM susceptibility, the most severe lung structure changes and lung function decline was observed in lung mesenchymal Tsc2-deficient breeding females.”
- Line 33 (now line 345) Other phase 1 LAM trials mentioned including the simvastatin and hydroxychloroquine trial with added reference 58.
- Line 332 (now line 378) changed sclerotic to sclerosis.
- Line 337-386 (now start at line 475) Do not add much to the review and may be biased by COI.
We changed “Another potential non-invasive biomarker for LAM is the carbohydrate-binding secreted protein Galectin-3 (Gal-3). Serum concentrations of Gal-3 have been shown to inversely correlate with Forced Expiratory Volume in 1 second (FEV1) in LAM patients not treated with Sirolimus [76]. Preclinical studies showed in global mesenchymal progenitor Tsc2 knockout tumors, with vascular anomalies, Gal-3 was overexpressed [77] and in Gal-3 knockout mice active β-catenin was reduced in the lung. These studies indicate that Gal-3 is related to both mTOR and Wnt pathway activation. Interestingly, there is an inhaled small-molecule Gal-3 inhibitor in Phase II clinical trials idiopathic pulmonary fibrosis (IPF) [78].
Changed to shorter comment and removed all comments on the Gal-3 inhibitor.
“Another potential non-invasive biomarker for LAM is the carbohydrate-binding secreted protein Galectin-3 (Gal-3)[76-78]. Preclinical studies showed in global mesenchymal progenitor Tsc2 knockout tumors, with vascular anomalies, Gal-3 was overexpressed [77] and in Gal-3 knockout mice active β-catenin was reduced in the lung.
